# Fluorine-Free Compound Water- and Oil-Repellent: Preparation and Its Application in Molded Pulp

**Xin Weng** [1], **Na Lin** [1], **Wenting Huang** [1] **and Minghua Liu** [1,2,*]

---

1   Fujian Provincial Engineering Research Center of Rural Waste Recycling Technology, College of Environment and Safety Engineering, Fuzhou University, Fuzhou 350116, China; 210620030@fzu.edu.cn (X.W.); bob3723@163.com (N.L.); 200620044@fzu.edu.cn (W.H.)

2   College of Environmental and Biological Engineering, Putian University, No. 1133, Xueyuan Middle Street, Chengxiang District, Putian 351100, China

*   Correspondence: mhliu2000@fzu.edu.cn; Tel.: +86-13305022089

**Abstract:** Molded pulp is considered an alternative to plastic packaging for its low cost, recyclability and non-pollution characteristics. However, the range of its applications has been limited by hydrophilicity and lipophilicity. Presented herein is a facile and straightforward method for the preparation biodegradable water- and oil-repellant for molded pulp. Sodium alginate-based oil repellent and PDMS-based water repellent were prepared by cross-linking and modification. The two were then mixed in various ratios to obtain compound water- and oil-repellent, which was applied to the molded pulp by dip-coating. The coated paper demonstrated excellent oil resistance (with a kit rating of 11/12) and outstanding water resistance (with a water contact angle of 121.9° and water absorption of 25.8%). This novel, eco-friendly, water- and oil-resistant molded pulp coating prepared from biodegradable and food-contactable materials is a potential candidate to replace petroleum-based coatings and has excellent possibilities to be manufactured on a large-scale intended for food and non-food contact applications.

**Keywords:** water resistant; oil resistant; molded pulp coating; fluorine-free; sustainable packaging; barrier property

## 1. Introduction

Packaging is vital in the supply chain system of procurement, production, transportation, sales and storage, which is closely related to human daily life [1]. As one of the four primary packaging materials, plastic products are universally utilized in the packaging industry due to their outstanding barrier properties, excellent mechanical properties and low prices [2,3]. However, plastic products are difficult to degrade in the natural environment [4], and their recycling is limited due to technical and economic factors. According to incomplete statistics, less than 3% of plastics are recycled in the world [5]. Therefore, there is an urgent need to find biodegradable packaging to replace plastic [6].

In this context, paper-based packaging stands out for its light weight, low cost and biodegradability [7–9]. Many kinds of paper-based packaging, including molded pulp, are typically made from recycled paperboard and/or newsprint [10]. Molded pulp products have been used extensively in the disposables market as the most promising alternative to plastic packaging [11]. However, molded pulp is made up of natural fibers with a porous and sparse structure causing it to be highly hygroscopic [12]. Compared with plastic materials, the molded pulp has a lower resistance to water vapor, water and grease [13].

Lamination and coating are the most commonly employed commercial approaches for providing water and oil repellency to paper products [14]. Low-density polyethylene (LDPE) liners are usually used in lamination treatment. However, the lining and paper fibers will be tightly bonded together after lamination, which is hard to separate and has a high recovery cost [15]. Therefore, laminated paper usually ends up in landfills as

waste [16]. Currently, coating is the most common method to improve the waterproof and oil-resistant properties of paper. Perfluoroalkyl compounds are considered to be the most widely used water- and oil-repellents [17]. However, it has been found that perfluorooctane sulfonate (PFOS) and perfluorooctanoic acid (PFOA) monomers used in the preparation of perfluoroalkyl compounds are critical organic pollutants with high persistence and bioaccumulation in the environment, which can increase the risk of various diseases and cause harm to humans [18–21].

Recently, research on coatings has shifted to the development of renewable and biodegradable biopolymer materials that have great water- and oil-repellency [22–24]. Polylactic acid (PLA) is a new type of biological degradable material prepared from starch that is extracted from renewable plant resources, such as corn. The FDA has recognized it as a suitable packaging material for all food products [25,26]. However, it is hard to detach PLA from the paper substrate in the pulping process as with LDPE, and it can only be degraded under composting conditions [24].

In addition, starch [27,28], chitosan [29,30], nanocellulose [31,32] and protein [33] have successfully caught the attention of researchers and paper industries for the preparation of water- and oil-repellent [8]. Dhwani Kansal et al. [28] coated the kraft paper with starch and corn protein separately, and the coated paper demonstrated good tolerance to oil and water, resulting in a Cobb 60 value of 4.81 $g \cdot m^{-2}$ and a kit rating of 12/12. Research by Zhao et al. [34,35] reported that the kit rating of chitosan-coated paper was 8/12 at a coating load of 9.33 $g \cdot m^{-2}$. After chemical modification with polydimethylsiloxane and castor oil, the oil resistance of chitosan-coated paper was improved, with a kit rating of 12/12 and 10/12 when coated with 10.33 $g \cdot m^{-2}$ and 4.3 $g \cdot m^{-2}$ paper, respectively. Xie et al. [36] proposed to coat the cellophane base paper with a fluorine-free cellulose coating, and the coated paper exhibited good barrier properties and mechanical properties. In general, the majority studies in order to achieve great water or/and oil resistance usually increase the amount of coating, which will inevitably increase the production cost of coated paper. In addition, there is a lack of research on biomass-based water- and oil-repellents that can be applied to pulp molding.

In this work, we proposed a facile method to fabricate an environmentally, economical and water- and oil-resistant molded pulp coating mostly using sodium alginate (SA) and polydimethylsiloxane (PDMS). Extracted from brown algae, sodium alginate has strong film-forming properties. The film has a strong barrier effect on oil and water vapor. PDMS has a low surface energy and a strong hydrophobicity as an inert material while still acceptable for food contact applications. However, the surface energy of PDMS is not that low-enough to block the penetration of oil into the porous paper [34]. In this study, SA and Polyethylene glycol (PEG) were produced as oil repellent under esterification reaction. PDMS was modified to prepare the water repellent. Then the water repellent and oil repellent were mixed in different volume ratios to derive the compound water- and oil-repellent for the coating of molded pulp. The fabricated SP/VAPDMS coated paper possessed excellent oil and water resistance and was able to wash off the coating completely after a simple pulp-washing procedure, which facilitated its application in sustainable development.

## 2. Materials and Methods

### 2.1. Materials

Polyethylene glycol (PEG, Mn = 1000) and p-Toluenesulfonic acid monohydrate (p-TSA) with 98% purity were obtained from Aladdin Co., Ltd. (Shanghai, China). Food grade sodium alginate (SA) was purchased from Mingyue Group Co. (Shandong, China). hydroxyl-terminated polydimethylsiloxane (PDMS-OH, 40 mpa·s), vinyltrimethoxysilane (VTMS, >98%), (3-Aminopropyl)triethoxysilane (APTES, >98%) and Ethanol absolute were obtained from Aladdin Co., Ltd. (Shanghai, China). Except for sodium alginate, all chemicals referred to are analytical grade. Molded paper was obtained from Ali-benben paper products Co., Ltd. (Shandong, China).

### 2.2. Preparation of SA-PEG Oil Repellent

To prepare a sodium alginate solution at a concentration of 3% (*w/v*), 1.5 g SA was added to 50 mL distilled water and mixed until completely dissolved. Added 0.13 g of p-TSA to the SA solution and stirred well. The stirred solution was transferred to a three-neck flask, heated at 50 °C and 135 r/min for 10 min, and then 0.5 g of polyethylene glycol was added and the temperature was raised to 60 °C for 10 h. The reactants were cooled to room temperature to obtain the SA-PEG oil repellent (SP).

### 2.3. Preparation of Water Repellent

To prepare a PDMS-OH solution at a concentration of 10% (*w/v*), 0.7 g of PDMS-OH was stirred with 7 mL of 95% (*v/v*) ethanol solution and stirred at 400 r/min for 20 min at room temperature. Then, added 0.6 g VTMS and 0.5 g APTES drop by drop into the PDMS-OH solution and stirred continuously for 3 h at 250 r/min to obtain water repellent (VAPDMS).

### 2.4. Preparation of Compound Water- and Oil-Repellent

Slowly added the VAPDMS into SP, mixed it in the volume ratio of 90:10, 80:20, 70:30 and 60:40, and stirred it at 25 °C and 300 r/min for 3 h. After removing impurities, compound water- and oil-repellent (SP/VAPDMS) was obtained.

### 2.5. Fabrication of SP/VAPDMS Coated Paper

Molded pulp samples were trimmed to 20 mm × 20 mm and weighed ($w_0$). The paper samples were then dipped into SP/VAPDMS solution for 1 min and then dried thoroughly in an oven at 60 °C, followed by pretreatment at 23 °C and 50% relative humidity (RH) and weighed again ($w_1$). The resulting SP/VAPDMS paper was obtained and its dip-coating load was calculated by Equation (1).

$$Dip - coating\ load\ \ (\%) = \frac{w_1 - w_0}{w_0} \times 100\% \tag{1}$$

### 2.6. Scanning Electron Microscopy (SEM)

A scanning Electron Microscopy (SUPRA 55, Carl Zeiss AG, Oberkochen, Baden-Württemberg, Germany) was used for imaging base paper and coated paper samples. Before SEM analysis, gold layer (15 nm) was coated on each sample by sputtering.

### 2.7. Fourier Transform Infrared (FTIR) Spectra

An ATR-FTIR spectrometer (AVAT-AR360, Thermo Nicolet Co., Madison, WI, USA) was used for the ATR-FTIR recording. 36 scans were performed in the spectral range of 4000–400 cm$^{-1}$ to derive the spectrum.

### 2.8. Thermogravimetric Analysis (TGA)

The thermal stability test was performed on base paper and coated paper using a thermogravimetric analyzer (DSC214, Netzsch Instruments, Germany). The paper samples were heated within the temperature range of 30–600 °C with a heating rate of 10 °C/min to obtain TG and DTG curves.

### 2.9. Oil Resistance

Oil resistance was tested on base and coated paper samples according to an international test method, the TAPPI standard T 559 pm-96. The kit rating solutions by mixing different ratios of castor oil, heptane and toluene to obtain 12 liquids with different surface tensions. A drop of the solution was dropped onto the paper sample and wiped off with a cotton ball after 15 s. The paper sample was then inspected for signs or spots left by the droplets. The larger the kit rating number, the more resistant the sample is to oil. The data were obtained as the average of the experiments conducted with three samples.

### 2.10. High-Temperature Oil Resistance

Tissue was placed on the back of the coated paper (10 cm × 10 cm), and 1–2 mL of 82 °C soybean oil was dropped onto the surface of the paper with a pipette. After 10 min, the excess oil was wiped off with a cotton ball. The samples were considered to block the hot oil if no oil penetrated the paper towel.

### 2.11. Water Absorption

The coated paper samples were dried at 70 °C for 1 h and weighed ($w_0$) before testing. The paper was then immersed in deionized water and weighed ($w_1$) for each sample at 0.5, 1, 2, 3, 5 and 24 h after removing excess water with a clean tissue. The water absorption rate can be obtained from Equation (2).

$$Water\ absorption\ (\%) = \frac{w_1 - w_0}{w_0} \times 100\% \tag{2}$$

### 2.12. Water Vapor Transmittance (WVTR)

The WVTR of the paper samples were measured in $g \cdot (m^2 \cdot 24\ h)^{-1}$ by the Permatran-W (model 3/34, Mocon Inc., Minneapolis, MN, USA) system at 23 °C and 50% RH. Trimmed to 2.54 × 2.54 $cm^2$, the paper samples were placed on an aluminum plate with a circular opening of 5 mm diameter in the middle. Prior to characterization, the samples were pretreated with nitrogen for 1 h under testing conditions.

### 2.13. Mechanical Testing of Tensile Strength

The molded pulp was cut to 50 mm × 15 mm and tested on a universal testing machine (model 1185, Instron Limited, Wycombe, UK) at a speed of 10 mm/min for tensile testing, following the ISO 527-2, 1993 (E) standard.

### 2.14. Contact Angles (CAs)

Water and oil contact angles were measured using an OSA200-01 system (Optical Surface Analyzer, Ningbo NB Scientific Instruments Co., Ningbo, China). A 5 μL droplet was dropped onto the paper and the contact angle was measured after 0 s, 30 s and 5 min. Three replicates of each paper substrate were taken and the average value was used as the result.

### 2.15. Repulpability

Paper samples (base and coated paper) cut to 2 cm × 2 cm were immersed in warm water (50 °C) for 60 min, and then divided into four portions after being pulpized using a blender. Wash the first three portions with deionized water, 2% (*v/v*) acetic acid solution and 85% (*v/v*) ethanol solution, respectively. The fourth part was washed with 2% acetic acid solution and then 85% ethanol solution. Finally, the above samples were washed with distilled water to eliminate the residual solvent. The four pulp parts were drained and then flattened with a hot iron and dried at 70° for 2 h to obtain the recycled molded pulps. Subsequently, the recycled molded pulps were analyzed by ATR-FTIR spectroscopy to compare the characteristic peaks for exploring whether the coating had been removed.

## 3. Results

### 3.1. Fourier Transform Infrared (FTIR) Spectra

FTIR spectra were used to verify the cross-linking reactions of SP oil repellent and modification of VAPDMS water repellent (Figure 1). As can be seen from Figure 1a, the SP paper has a characteristic peak of −OH at 3300–3700 $cm^{-1}$ compared to the base paper and the peak here is broadened, probably due to the overlap of the unreacted carboxyl and hydroxyl peaks in the alginate molecule. The sharp peak around 1601 $cm^{-1}$ is attributed to the C=O peak generated by the esterification reaction between polyethylene glycol and alginate. Since only some of the carboxyl groups reacted with the hydroxyl group,

the coexistence of carboxyl groups and ester bonds in the reaction product led to the overlapping of the C=O peaks of the two, resulting in a relatively large change in the width of the peak. In addition, a more obvious C−O stretching vibration peak appeared at 1086 cm$^{-1}$, which is a more obvious characteristic peak of the ester group. The above analysis results indicate that the chemical reaction between SA and PEG has occurred, and the polyethylene glycol-based oil repellent has been successfully prepared in this experiment. From Figure 1b, it can be seen that VAPDMS paper has a characteristic peak of Si−O−CH$_3$ at 2845–2975 cm$^{-1}$ compared to base paper, and the −CH$_3$ group is considered to be a group with low surface energy [37]. Moreover, the presence of Si−CH$_3$ at 799 cm$^{-1}$ confirms the presence of PDMS. The modified PDMS introduced C=C and C−N, so that the stretching vibration peaks of vinyl C=C and C−N appeared near 1590 cm$^{-1}$ and 1260 cm$^{-1}$. These four characteristic absorption peaks were not seen in the base paper, but the modified PDMS has these three characteristic absorption peaks, which can be judged that VTMOS and APTES have been successfully attached to PDMS, and the hydrolytic condensation reaction between PDMS and VTMOS and APTES has occurred, and the polydimethylsiloxane-based water repellent (VAPDMS) has been successfully prepared.

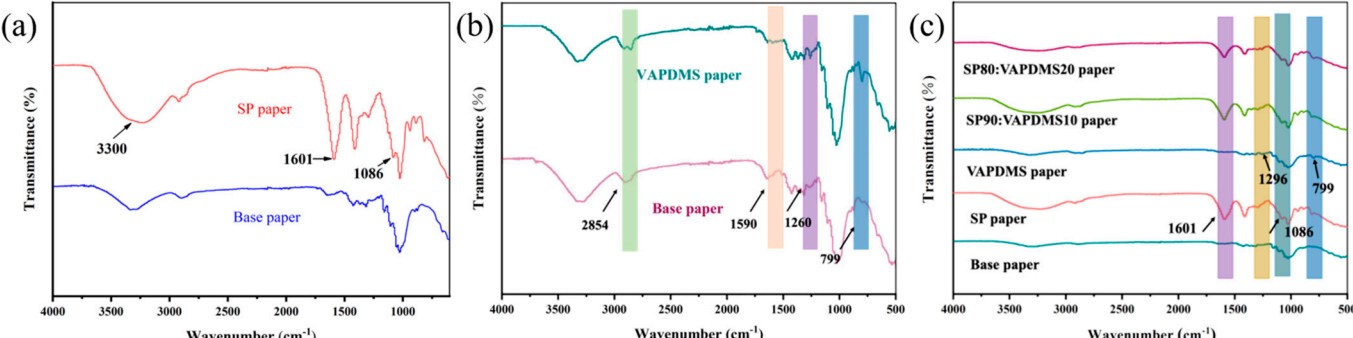

**Figure 1.** (**a**) FT−IR spectra of base paper and SP paper; (**b**) FT−IR spectra of base paper and VAPDMS paper; (**c**) FT−IR spectra of base paper, SP paper, VAPDMS paper, SP90:VAPDMS10 paper and SP80:VAPDMS20 paper.

Figure 1c demonstrates the FT−IR spectra of base paper, as well as paper samples coated with oil repellent (SP), water repellent (VAPDMS), and compound water- and oil-repellent (SA/VAPDMS). SA/VAPDMS paper had the characteristics of SP and VAPDMS papers peaks. Still, its infrared spectral characteristics were closer to those of SP due to its higher ratio. What's more, SP90:VAPDMS10 paper was more relative to SP than those of SP80:VAPDMS20. In addition, the intensity of the characteristic peaks of the O−H bond of SA/VAPDMS was significantly weaker, and the peak shape was wider compared with that of SP and VAPDMS, indicating the existence of a relatively strong hydrogen bonding between the water repellent and the oil repellent.

*3.2. SEM*

In order to understand the barrier mechanism of the coated paper at the microscopic level, SEM images of base paper, SP90:VAPDMS10 paper, SP80:VAPDMS20 paper and SP70:VAPDMS30 paper were recorded (Figure 2). It can be clearly observed from the SEM image of Figure 2a that the virgin paper fibers were mis-woven and had significant pores, which could allow the passage of water and oil. Figure 2b–d showed SEM images of the SP/VAPDMS paper with a film formed on the surface, which masked most of the pores. Furthermore, it can be observed that the higher percentage of SP in compound water- and oil-repellent, the better coverage and smoothness it would have.

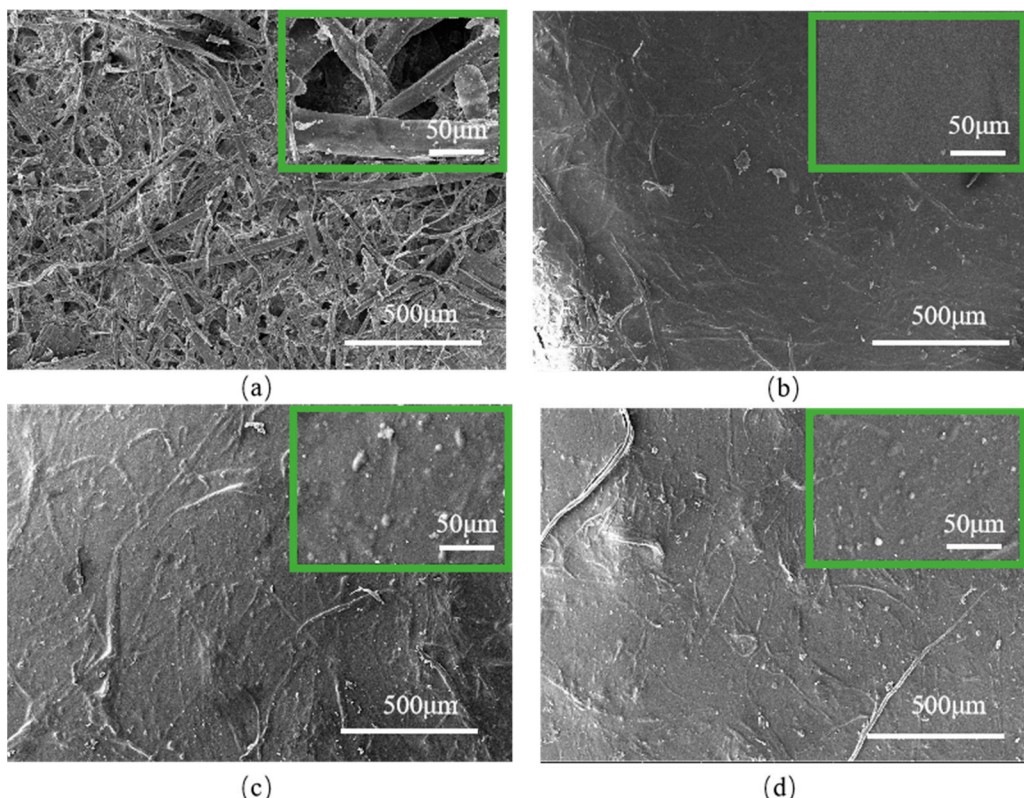

**Figure 2.** SEM images (×100) with zoomed-in pictures as insets (×1000) in the upper right corners of base paper (**a**), SP90:VAPDMS10 paper (**b**), SP80:VAPDMS20 paper (**c**) and SP70:VAPDMS30 paper (**d**).

### 3.3. Thermogravimetric Analysis (TGA)

The thermal stability of the coating layer was investigated by thermogravimetric analysis (TGA). TG and DTG curves of base paper and SP/VAPDMS paper were shown in Figure 3. Since the coating load (5.3%) was small, the overall trend of the TGA curves for the base and coated papers was not significantly different. It was analyzed that the mass loss observed below 120 °C corresponded to the evaporation of water from the paper. In the range of 250 °C to 375 °C, the significant mass loss was attributed to the decomposition of the paper and coating material. After 480 °C, the SP/VAPDMS paper lost less mass than the base paper, which could be due to the enhanced thermal stability of the compound water- and oil-repellant. In conclusion, as coated paper has no noticeable mass loss below 200 °C, it indicated that the coated paper was thermally stable for high temperature applications.

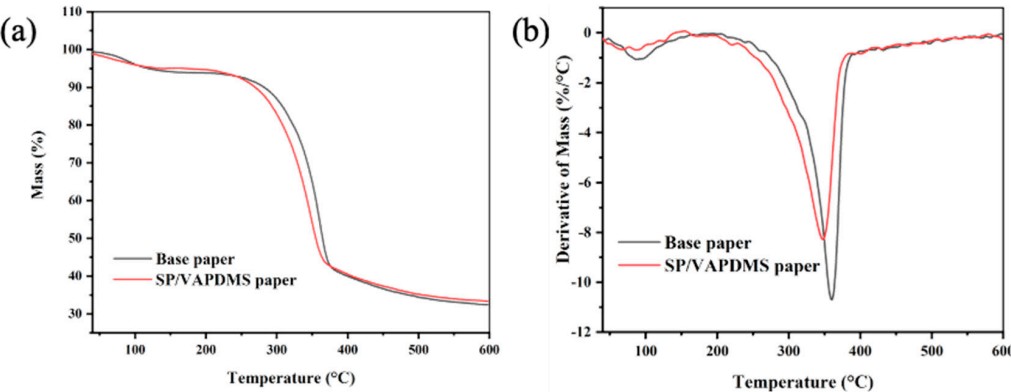

**Figure 3.** The (**a**) TG and (**b**) DTG curves of base paper and SP70:VAPDMS30 paper.

### 3.4. Oil Resistance

The oil resistance of SP/VAPDMS was investigated by dip-coating different loads of SP70:VAPDMS30 on the paper, and the results were illustrated in Figure 4a. The base had a kit rating of zero, indicating no oil resistance. The kit rating for 9.6% dip-coating load coated paper was 5/12. It met the requirements of greaseproof paper with a kit rating higher than or equal to 5 according to the TAPPI standard [38]. The kit rating for 20.9% dip-coating load coated paper was 11/12. It can be seen from the data that the dip-coating load improved the kit rating very significantly. Due to increased dip-coating load, the coating thickness will increase, which can better cover the paper fiber pores to prevent grease penetration. Therefore, the coating's integrity, uniformity, and compactness of the coating had an important role to play in the oil repellency of the paper [39,40].

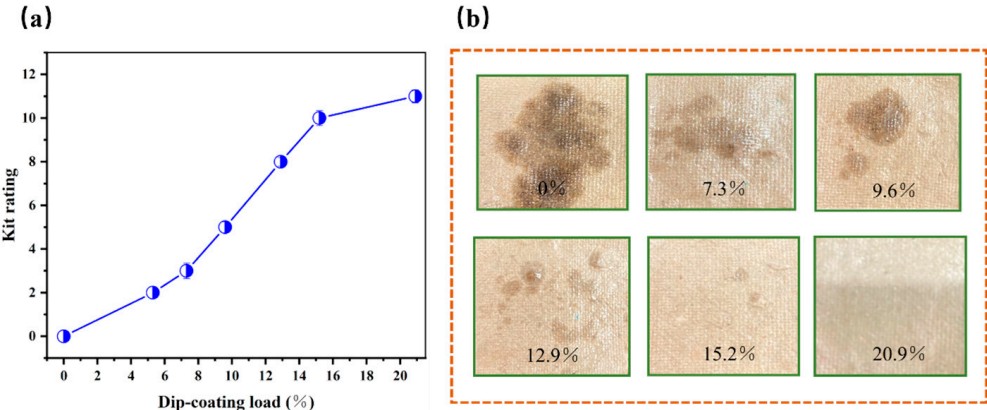

**Figure 4.** (**a**) The influence of dip-coating load on the oil repellency of SP/VAPDMS paper; (**b**) Thermal oil resistance testing of SP/VAPDMS paper. x% represents the load of SP70:VAPDMS30 on paper.

### 3.5. Thermal Oil Resistance

Food packaging will inevitably encounter high-temperature food in daily use, so the resistance of molded pulp to high-temperature grease significantly impacts its applicability. In this regard, we tested the paper against thermal oil at different dip-coating loads (Figure 4b). When the dip-coating load was less than 12.9%, the oil droplets penetrated the paper, and the dark spots on the surface were prominent. When the dip-coating load was 15.2%, the area of dark spot area was significantly reduced, and only a tiny amount of oil drops penetrated the paper. There were no black spots when the dip-coating load was increased to 20.9% and the paper achieved the best high-temperature grease resistance.

### 3.6. Water Absorption

The water repellency of SP/VAPDMS paper could be demonstrated by the water absorption rate (Figure 5a). The water absorption rate of the base paper was 93.1% at the beginning of 0.5 h and reached 136% after 24 h. The SP/VAPDMS paper showed remarkably low water absorption compared to the base paper, and the water absorption decreased as the proportion of water repellent increased. SP60:VAPDMS40 demonstrated a water absorption of 25.8% after 24 h, which was about 110% lower than the base paper. It was due to the hydrophobic group in the water repellent that allowed the paper to obtain low surface energy [41], thus considerably reducing hydrophilicity.

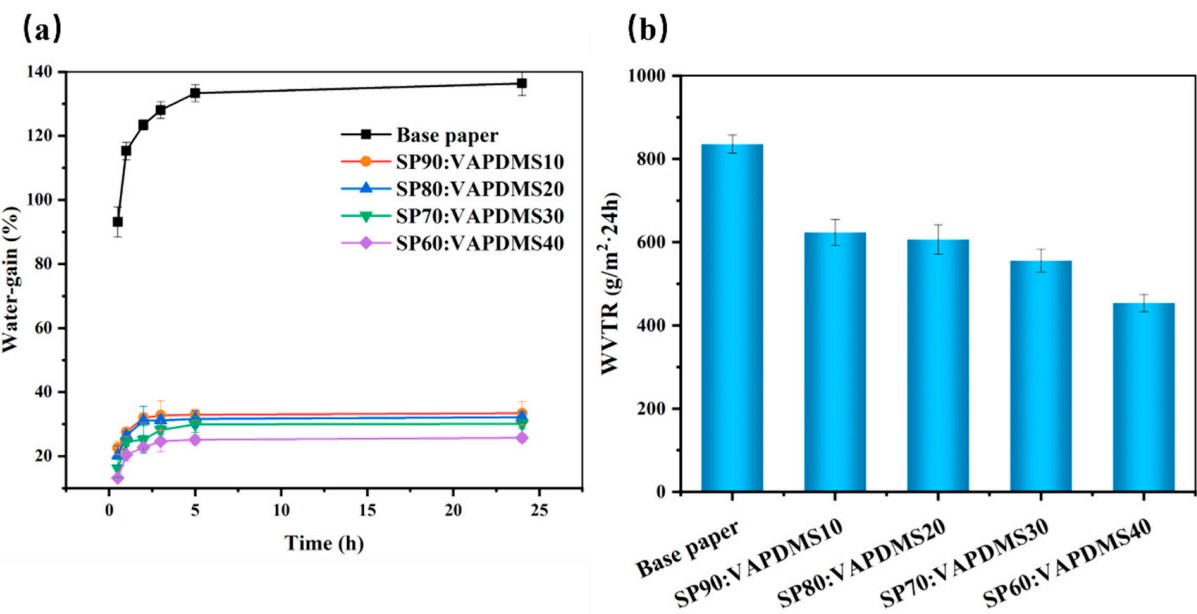

**Figure 5.** (**a**) Water absorption of base paper, SP90:VAPDMS10, SP80:VAPDMS20, SP70:VAPDMS30 and SP60:VAPDMS40 paper; (**b**) WVTR (g·(m²·24 h)$^{-1}$) of base paper, SP90:VAPDMS10, SP80:VAPDMS20, SP70:VAPDMS30 and SP60:VAPDMS40 papers.

*3.7. Water Vapor Transmittance (WVTR)*

The WVTR of base and coated paper were also tested with a desiccant method (Figure 5b). The water vapor permeability of virgin paper was large, however, the permeability of SP/VAPDMS paper was greatly reduced compared to base paper. This was observed due to the hydrophobic nature of VAPDMS and the coverage of SA, which resulted in lower water vapor transmission and led to lower water vapor permeability. In addition, the increasing proportion of VAPDMS in the SP/VAPDMS contributed to a decrease in water vapor permeability. The lowest WVTR at 454 g·(m²·24 h)$^{-1}$ was found in SP60:VAPDMS40 paper, which represented a 46% decline compared to 836 g·(m²·24 h)$^{-1}$ for base paper. Overall, the content of the water repellent affected the water vapor barrier properties. The hydrophobic functional groups attached to the paper surface and the film formed by the oil repellent worked together to block the entry of water vapor.

*3.8. Water/oil Contact Angles*

To evaluate the wettability of base paper and coated paper to water and castor oil, an optical surface analyzer was used to test their water and oil contact angles at 0 s, 30 s and 5 min (Figure 6a,b). The base paper demonstrated moderate resistance to wetting at 0 s and 30 s, but as time elapsed, water completely penetrated the base paper after 5 min, with a WCA of 0°. SP paper was barely waterproof, the WCA at 0 s was found to be 21.3 ± 1.8°. VAPDM paper had the greatest wettability resistance, and the WCA was always stabilized at about 150 ± 3.1°. For different volume ratios of SP/VAPDMS (90:10, 80:20, 70:30, 60:40), the WCA increased with the proportion of water repellent. The SP90:VAPDMS10 paper showed a WCA of 56.4 ± 3.5° at 30 s, while the SP60:VAPDMS40 paper showed a WCA of 121.3 ± 1.6° at 30 s. The decreasing order of WCA was VAPDMS paper > SP/VAPDMS paper > Base paper > SP paper. In summary, the hydrophobicity of the coated paper depended mainly on the content of the waterproofing agent. The percentage of water repellents in the compound water-and oil-repellents increased, causing the hydrophobicity to increase. Still, the presence of hydrophilic oil repellents led to a lower WCA than VAPDMS. From the figures, it was evident that the WCA of SP70:VAPDMS30 and SP60:VAPDMS40 were stabilized and were more than 90° even after 5 min. For manufacturing budget reasons, the SP70:VAPDMS30 was chosen to validate its practical application. As shown in Figure 6c, water penetrated the base paper after 5 min,

while the SP70:VAPDMS30 paper blocked the permeation of water, showing excellent water repellency.

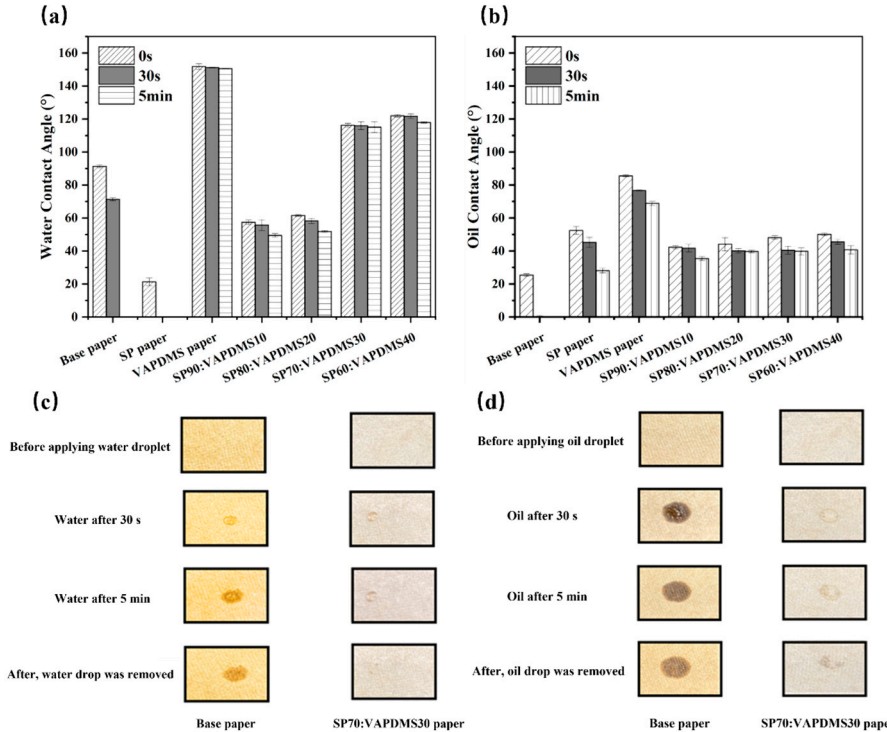

**Figure 6.** (**a**) Water contact angles (WCA) for base paper SP90:VAPDMS10 paper, SP80:VAPDMS20 paper, SP70:VAPDMS30 paper, and SP60:VAPDMS40 paper for 0 s, 30 s, and 5 min, respectively; (**b**) Oil contact angles (OCA) for base paper SP90:VAPDMS10 paper, SP80:VAPDMS20 paper, SP70:VAPDMS30 paper and SP60:VAPDMS40 paper for 0 s, 30 s, and 5 min, respectively; (**c**) Water droplet behavior on the base paper and SP70:VAPDMS30 paper, respectively; (**d**) Oil droplet behavior on the base paper and SP70:VAPDMS30 paper, respectively.

The oil contact angle (OCA) of base paper and coated paper, was shown in Figure 6b. An oil contact angle of 0° after 30 s was observed for the base paper, which suggested that it had no oil resistance. VAPDMS exhibited the highest oil contact angle, however, when the surface oil droplets were wiped away, the VAPDMS paper surface darkened over a large area, indicating that it was not impermeable to oil. For different volume ratios of SP/VAPDMS (90:10, 80:20, 70:30, 60:40), the oil contact angles were similar and stabilized at about 40° after 5 min. Compared with SP paper and VAPDMS paper, the OCA of SP/VAPDMS paper was lower than theirs, which could be due to the presence of the more reactive polar groups when SP and VAPDMS were mixed [42]. The SP70:VAPDMS30 was used to demonstrate its practical application, as shown in Figure 6d. The oil had wholly penetrated the base paper after 30 s, while the SP70:VAPDMS30 paper prevented the oil from penetrating. When the oil droplets were wiped off after 5 min, there were no areas of darkening on the surface, indicating that it had the potential to be used in food packaging with good oil repellency.

*3.9. The Mechanical Properties*

The assessment of the mechanical properties of the paper sample was shown in Figure 7. The tensile strength of the base paper was poor because the paper was a network structure consisting of natural fibers connected by hydrogen bonds, which can easily crack when subjected to mechanical stress. The coating on the surface of SP paper is due to the evaporation of water after drying to form a continuous dense film making the molecules of the substance closer to each other, which in turn increases the tensile strength of the food-contactable packaging. In addition, the coating not only forms a film on the surface of

the paper, but also partially penetrates into the pores, strengthening the bond between the paper fibers and ensuring the quality of the packaging material. There is a difference in the tensile strength of base paper compared to VAPDMS paper, which is mainly related to the strength of hydrogen bonds between paper fibers. The higher stress in the VAPDMS paper is attributed to the fact that the addition of the water repellent makes the internal structure of the molded pulp change and strengthens the bonds with hydrogen [43], so the tensile strength of the VAPDMS paper is further improved.

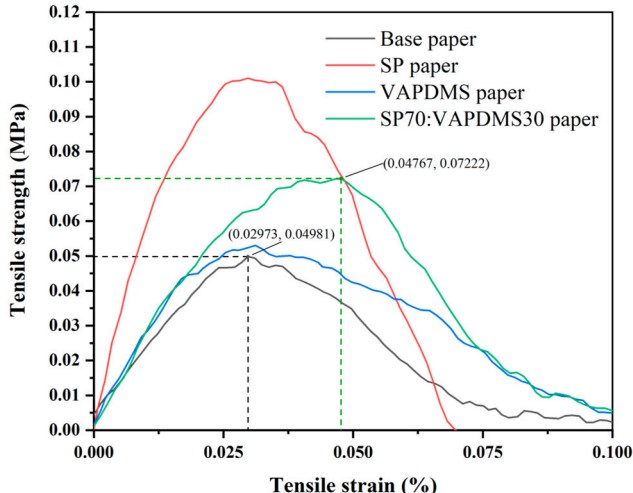

**Figure 7.** The stress and strain of base paper, SP paper, VAPDMS paper and SP70:VAPDMS30 paper.

Compared to base paper, SP70:VAPDMS30 paper exhibited significant tensile properties with a tensile strength of 0.07222 MPa, an improvement of 45%. Since the volume ratio of oil repellent in SP70:VAPDMS30 paper occupies much, the curve is similar to the stress-strain curve in SP paper, which fully indicates that the mixing of water and oil repellent does not reduce its tensile performance and can be applied to the packaging industry as well. In addition, the tensile curve of paper molded pulp showed multiple wave peaks, the possible reason for this is that paper molded pulp is a fiber network structure with a certain thickness, and the fracture is not completed in an instant, but the process of constantly breaking the internal hydrogen bonding force.

### 3.10. Repulpability

The most difficult part of the pulp recycling process is the separation of the pulp from the coating, especially when the coating is made of synthetic polymers. As a result, coated paper is generally disposed in landfills, which results a waste of resources and may harm the ecosystem. Since the coating materials are mainly manufactured from eco-friendly materials, it is possible to recycle pulp by separating the coating from the paper as needed. Considering these factors, we washed the dip-coated paper with several solvents and successfully removed the SP/VAPDMS coating from the pulp. From Figure 1c, it can be seen that VAPDMS paper introduced C=C and C−N, so that the stretching vibration peaks of vinyl C=C and C−N appeared near 1590 cm$^{-1}$ and 1260 cm$^{-1}$. Comparing the infrared spectrogram of the water- and oil-repellant with that of the water repellant, it can be seen that the water- and oil-repellant appears C=C and C−N. The C=C of the water- and oil-repellant appears at 1601 cm$^{-1}$, and the wavenumber moves to a high wavenumber compared with the C=C of the water repellant, and the intensity of the absorption peak is weakened, indicating the existence of hydrogen bonding between the water repellant and oil repellant. Water- and oil-repellant at 1296 cm$^{-1}$ carbon and nitrogen characteristic absorption peak compared with the water repellant in the carbon and nitrogen to move to a high wavenumber, and absorption intensity is reduced, indicating that the carbon and nitrogen bonds involved in the formation of intermolecular hydrogen bonding. The Si−CH$_3$ at 799 cm$^{-1}$ of the water- and oil-repellant coincides with the characteristic peak

in the water repellent, so it is not involved in the formation of intermolecular hydrogen bonds. Comparison of the ATR-FTIR spectra (Figure 8) of washed and unwashed recycled paper was used to detect the presence of residual coatings in the pulp. The spectra of the unwashed repulped paper were compared with those of the coated repulped paper washed with ethanol and acetic acid. After washing coated paper pulp with 85% ethanol aqueous (E), the peaks corresponding to ester groups (1601 cm$^{-1}$ and 1086 cm$^{-1}$) and C$-$N, Si$-$CH$_3$ (1296 cm$^{-1}$ and 799 cm$^{-1}$) disappeared, and the infrared spectrum were the same as that of base paper, which confirmed that the coating was basically washed off.

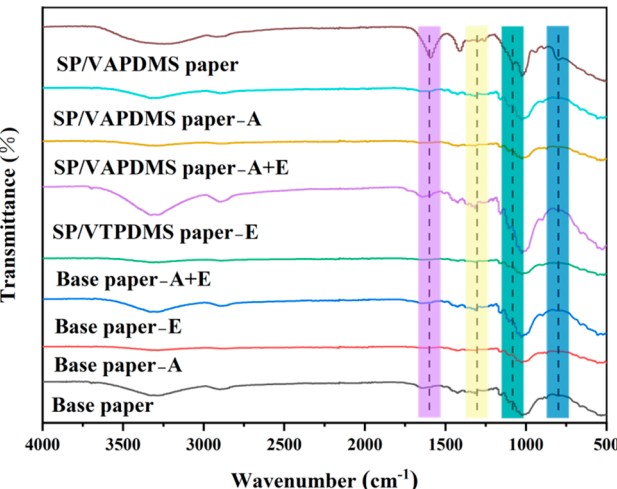

**Figure 8.** FTIR spectra of repulped papers made from fibers of base paper and SP70:VAPDMS30 paper, with 2% (*v/v*) acetic acid solution, 85% (*v/v*) ethanol solution and water.

## 4. Conclusions

In conclusion, we have developed a novel method to create an environmentally friendly water- and oil-resistance paper. The coated paper demonstrated excellent oil resistance (with a kit rating of 11/12) and outstanding water resistance (with a water contact angle of 121.9° and water absorption of 25.8%). The water vapor barrier performance of the coated paper was significantly improved, by 48.1% relative to the base paper. SEM analysis confirmed the disappearance of pores on the paper fiber surface after dip-coating. In addition, the coated paper exhibited good thermal stability and excellent mechanical properties. After washing and repulping tests, ATR-FTIR spectra of recycled paper demonstrated the recyclability of the pulp. This food-safe paper coating using environmentally friendly, biodegradable ingredients can be used to replace fluorinated coatings in the packaging industry, which will offer corresponding economic and environmental benefits.

**Author Contributions:** Conceptualization, X.W.; formal analysis, N.L. and W.H.; investigation, X.W. and W.H.; supervision, M.L.; validation, N.L. and M.L.; writing—original draft preparation, X.W. and W.H.; writing—review and editing, X.W. and M.L. All authors have read and agreed to the published version of the manuscript.

**Funding:** This work was financially supported by the Key Projects of Science and Technology Innovation in Fujian Province, China (No. 2021G2001) and External cooperation projects (cooperation) of Fujian Provincial Science and Technology, China (No. 2020I0043).

**Institutional Review Board Statement:** Not applicable.

**Informed Consent Statement:** Not applicable.

**Data Availability Statement:** All data generated or analyzed during this study are included in the present article.

**Conflicts of Interest:** The authors declare no conflict of interest.

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
