# Peer review of "Fluorine-Free Compound Water- and Oil-Repellent: Preparation and Its Application in Molded Pulp"

_coatings, doi:10.3390/coatings13071257_

Round 1

Reviewer 1 Report

Dear authors,

The manuscript is prepared according to journal MDPI-Coatings. Aim of the research is in the scope of the mentioned journal. Article presents interesting and current topic of the preparation of Molded Pulp, using Fluorine-free Compound Water- and Oil-repellent. The aim of this work was to develop an environmentally friendly and cost-effective molded pulp coating using sodium alginate and polydimethylsiloxane, which have great potential as applications.

1 The abstract provides aims and important findings of the research.

2.      Introduction: Introduction provides sufficient background and includes all relevant references.

3.      Materials and methods: How the specimen were tested according to the chapter 2.13 Mechanical testing? The testing procedure employed for the specimens involved in this study was conducted according to the TAPPI standard. This standard entails the cutting of specimens from the designated materials, followed by the execution of tensile tests until the point of specimen failure is reached.

4.      Results: Could you please explain more in detail the findings in Figure 7, which display the results of mechanical tests like tensile strength and tensile strain?  At which tensile strength the rapture of the specimen occurred?

5.      Conclusions: provide appropriate and the most important findings of the research.

Reviewer 2 Report

This manuscript reports on the study of a novel, eco-friendly, water- and oil-resistant molded pulp coating prepared from biodegradable and food-contactable materials

The topic and the results are very interesting and the characterization is thorough. However, there are some parts to improve in the text which need to clarify.

 In particular,

1) The role of the reagents listed in the experimental part and the reactions between them should be better described in the results.

The authors mention crosslinking and ester formation which, in my opinion, are not detected in the spectroscopic characterization. What role do PEG, VTMS and APTES play? How were they selected and what role do they play in the final formulation? These reagents are indicated in the experimental part but then never commented on in the study of the formulations

2) Comparison of FT-IR spectra is not easily evaluated from the figure. Incorrect attributions are also reported in the text, such as the band at 1640 cm-1 which is too low to be an ester carbonyl. Furthermore, is the decrease in the band attributed to OH stretching due to the presence of hydrogen bonds or rather to a lower concentration of OH groups in the mixtures?

3) Could the behavior observed in the thermal analysis be attributed to a different percentage of inorganic residue in the treated paper? In fact it has been said that heating below that temperature leads to the degradation of all organic compounds.

4) line 339 Repulpability: The attribution of peak corresponding to ester groups at 1601 cm-1 is not correct and also the peaks attribution "C-N, Si-CH3 (1296 cm-1 and 799 cm-1)" should be written more clearly

5) The conclusions must be correct based on the previous comments

In conclusion, in my opinion, the authors should review the manuscript and some corrections are necessary for the publication on Coatings.

Round 2

Reviewer 2 Report

The authors responded to the observations, however only partially resolving some of the critical issues I highlighted:

Response 1: In my opinion, a description of the synthesis steps prior to the discussion of the spectra and other results would have improved the quality of the manuscript. The authors chose to augment the explanations in section 3.1 but they need to be well commented at this point.

Response 2 and 4: The authors explained in their review the position of the signal attributed to the ester group as deriving from an overlap of the signal with the non-esterified groups of the alginate. This assumption is correct but therefore does not allow the formation of esters groups to be clearly identified. In particular, in my opinion, the spectrum of SP shown in fig 2-a is different from the spectrum shown for SP in fig 2-c.

For the spectrum of fig 2-a the signals could be compatible with an excess of residual humidity in the sample. The signal attributable to the cellulosic skeleton, visible instead in the spectrum of fig 2-c, is almost not visible in 2-a. I think the authors need to check this inconsistency and clarify why there are two different spectra for SP

Given the importance that has been given by the authors to FT-IR spectroscopy to explain the reactions carried out, it is in my opinion essential to clarify these critical issues

Round 3

Reviewer 2 Report

Comments made on the previous version were received sufficiently to accept the manuscript for publication.